# Peer Support and Exercise Adherence in Adolescents: The Chain-Mediated Effects of Self-Efficacy and Self-Regulation

**DOI:** 10.3390/children10020401

**Published:** 2023-02-18

**Authors:** Yuxin Zou, Shijie Liu, Shuangshuang Guo, Qiuhao Zhao, Yujun Cai

**Affiliations:** 1School of Physical Education, Shanghai University of Sport, Shanghai 200438, China; 2School of Earth Sciences, Zhejiang University, Hangzhou 310058, China

**Keywords:** exercise adherence, peer support, self-efficacy, self-regulation, adolescent

## Abstract

In the context of exercise psychology, the mediating relationship between peer support, self-efficacy and self-regulation, and adolescents’ exercise adherence was to be explored. Methods: A questionnaire was distributed among 2200 teenagers from twelve middle schools in Shanghai. The “process” program in SPSS and the bootstrap method were applied to construct and analyze the direct and indirect effects of peer support on adolescents’ exercise adherence. Results: Peer support directly affected adolescents’ exercise adherence (β = 0.135, *p* < 0.001, effect size of 59%) and self-efficacy (β = 0.493, *p* < 0.001, effect size accounted for 42%), and self-regulation (β = −0.184, *p* < 0.001, effect size of 11%) influenced exercise adherence indirectly. In addition, self-efficacy and self-regulation could impose a chain-mediated effect on peer support and exercise adherence (effect size of 6%). Conclusion: Peer support could promote adolescents’ exercise adherence. Self-efficacy and self-regulation are mediating factors of peer support on exercise adherence in teenagers, self-regulation as well as self-efficacy-imposed chain-mediating effects on peer support and adolescents’ exercise adherence.

## 1. Introduction

It is universally acknowledged that sports activity is beneficial for both physical and mental health. However, only a few people adhere to long-term exercise [1]. The World Innovation Summit for Health released a report in 2022 [2], indicating that one in four adults as well as four in five teenagers do not get enough exercise in terms of intensity and time. Many people will give up physical exercise in a short time, even if they have an exercise plan. According to statistics from Sperandei’s team [3], 63% of adults who participated in physical activity quit the activity within a year. As modern technology continuously develops, this phenomenon is becoming increasingly worrying. For example, the emergence of mobile applications distract people’s attention and weaken their exercise durability.

Only by keeping to long-term workouts can we achieve the expected effect, while short-term exercise will not meet one’s ends. Changing one’s lifestyle, including long-term exercise, is challenging for people of all ages [4]. This is especially the case for teenagers, who are still young and experiencing the golden stage of physical growth. Children and adolescents with exercise habits benefit from physical and mental health, sleep quality, brain development, bone health and cognitive health, but only 20% of adolescents exercise for one hour per day [5]. Despite a number of studies [6] and policies indicating that teenagers should get more than 1 h of medium to high intensity physical activity every day, most quit after PE class because they are not only unaware of the importance of lifelong sport, but also they have not formed the habit of completing voluntary workouts during school time. However, exercise adherence could settle the issue once and for all, because external force can only influence one’s behavior partially, while the internal serves as the base to promote and maintain change [7]. Exercise adherence, a mental state in which exercisers constantly overcome difficulties and voluntarily participate in regular physical workouts for a long time so as to gain the ends of the former activity [2], refers to the sum of implicit performance (emotional experience, endeavor) and explicit performance (behavioral habits) [8]. It could also enhance one’s physical fitness, lead to effective and sustainable weight loss [9], meanwhile keeping a pleasant mood [10]. Therefore, by improving teenagers’ sports persistence, we can achieve the goal of making them exercise.

There are many scholars who have conducted relevant studies on the exercise adherence of adolescents, but most of the studies focus on obese children, diabetic children [11], children with musculoskeletal diseases [12], and other unique populations, studies on the exercise adherence of adolescents in the public domain are relatively few. However, it cannot be denied that exercise adherence has an impact on all adolescents. Therefore, this paper aims to study adolescents’ exercise adherence to help them develop voluntary workout behavior, hence improving their physical health.

Exercise adherence requires the exerciser to be physically active on a regular basis. Nowadays, more people realize the significance of peers on physical exercise. Discussions on exercise adherence should be based on analysis of exercise participation [13]. A large amount of the literature has found that adolescents with a high level of peer support tend to have a higher probability of participating in sufficient physical activity [14]. However, the research on exercise adherence needs to be improved. No single variable can explain and predict exercise adoption and adherence [15]. Basic psychology needs theory (BPNT) [16], as a branch of self-determination theory (SDT), holds that individuals generally have three basic demands: the demand for relatedness, the demand for autonomy, and the demand for competence. These three basic needs play a key role in human growth and are important in this article for adolescent exercise activities.

The need for relatedness is described as a sense of belonging and a connection to others. Each person has many needs for relationships, but because of their young age and little social contact, the need for parental and peer relationships is most important for adolescents. Peer relationships are not as close as parental relationships, but they are physically close. However, adolescents spend more time in school every day, so they are more likely to have closer contact with their peers, and they will not obey parental constraints during a rebellious period. Compared to emotional and psychological kinship relationships, adolescents identify with peer relationships no less than kinship relationships. Peer support is a process of sharing emotions, experiences and behavioral skills with those similar in age, life experience and living environment [17]. The existence of peer support explains the positive influence and role of peer relationships, and is considered to be one of the key factors influencing children and adolescents’ exercising behavior. In the study, Peers for Progress: promoting peer support for health around the world developed by the World Health Organization (WHO), a strategic approach was put forward to promote peer support for health worldwide [18]. As for improving exercise adherence at all ages, scholars have put forward many promotion methods, including peer support [19]. Studies have shown that peer support has a positive effect on adolescents’ physical exercise. When accompanied by a peer, adolescents have a higher chance of participating in physical exercise [20]. As adolescents grow up, considering their autonomy and growing peer relationships, the effect of parental support on children and adolescents gradually decreases, thus bringing about a greater impact from peer support [21]. Although mid-adolescence (14–16 years) is the period when adolescents are most influenced by their peers [22], early adolescence (11–13 years) is also influenced by peers in various ways, including exercise, education [23], etc. During this time, teenagers tend to spend more time socializing with their contemporaries (e.g., face-to-face or via SMS and social networks) [24]. Peer support provides adolescents with motivation and helps them make better choices about their lifestyle. For instance, exerting an effect on each other’s participation in physical activities [25], i.e., inspiring adolescents to consciously persist in exercising.

The demand for autonomy is an act of will, more specifically, a sense of will and choice when acting [26]. Human beings aspire to maintain a complete sense of self. It is manifested in our innate tendency towards self-management behavior [27]. Self-efficacy [28] refers to an individual’s belief in the ability required to perform a certain behavior. It is one of the ways in which the individual psychologically manages themselves. In other words, the sense of belief expressed by self-efficacy is a manifestation of the need for autonomy. When an individual feels that he or she is the initiator of their choices and decisions and behaves in accordance with his or her self-perception, his or her need for autonomy is satisfied. The satisfaction of autonomous needs usually leads to autonomous motivation and adaptive individual behaviors, such as consistent exercise. That is to say, self-efficacy is a part of the autonomy need, and when this part of the demand is met, self-efficacy can generate motivation for behavior and improve individuals’ capabilities for success. It is this internal drive that motivates teenagers to change their way of life and insist on physical exercise, which acts as an inexhaustible power. On the one hand, self-efficacy affects people’s choice of activities, the degree of effort and the degree of adherence to activities, which is manifested as the influence on exercise adherence. Multiple studies have shown that self-efficacy has an important impact on adolescents’ physical exercise behavior [29,30]. People with a strong sense of self-efficacy will show higher confidence when dealing with various challenges, so as to insist on their choice of behavior [31]. That is to say, the presence of self-efficacy makes people choose to continue exercising despite setbacks.

The need for ability is also known as the need for competence. Some scholars think that ability is related to a person’s performance ability in the environment [27], while others think that ability is related to whether a person is an effective actor in the environment [26]. Self-regulation [32] refers to a physiological process that psychologically enables individuals to guide their own target activities over time and space, as well as the ability to change their own behaviors [33]. Although it cannot be said that self-regulation is a competency need, the ability to self-regulate must be part of the competency need. Self-regulation mentally directs people to change their target activities, which is a manifestation of ability. This ability allows one to maintain an identity in a self-centered environment, which is the way to be an effective actor. In this case, it is possible to meet one’s own capacity requirements. People with high confidence in their emotional regulation can adopt effective strategies to adjust their emotions. There are also scholars who carry out research on emotional self-regulation. The emotional response under moderate exercise can increase the willingness to exercise in the future [34]. The choice of exercise is based on cognitive factors, such as weighing the benefits and disadvantages [35]. In addition, the decision on whether to insist on exercise is influenced by emotion, based on whether you had a pleasant workout last time [36].

When these needs of the people are satisfied, it is often accompanied by the creation of motivation. Deci and Ryan [37] believe that when the three psychological needs are met, people will actively engage in the environment and have a higher recognition of and identification with the values and culture around them, thus inspiring greater interest in exercise adherence. That is, when the relational, autonomy and competence needs are satisfied, motivation to persist in exercise can be promoted. Even though the three needs are not equivalent to peer support, self-efficacy, and self-regulation, they are all one of the manifestations of these three needs. When the demonstrated peer relationship, self-efficacy, and self-regulation play a positive role, it also means that the three needs are met to some extent. When someone is supported by peer relationships, self-efficacy and self-regulation are affected, which will produce the motivation for independent exercise, that is, enhance exercise persistence. Meanwhile, the more positive effects in Schutte’s review were related to the need for relationships; the desire for peer relationships may lead adolescents to be more active in exercise [38]. According to the points above, this study proposed the following hypothesis: H1: Peer support influenced adolescents’ exercise adherence as a direct precursor; H2: Self-efficacy and self-regulation play an independent mediating role in exercise adherence with regard to interpersonal relationships; H3: Self-efficacy and self-regulation play a chain-mediating role in the influence of exercise adherence. Generally, this study aims to explore the chain-mediation mechanism (self-efficacy and self-regulation) between peer support and exercise adherence (Figure 1), inspiring ideas to motivate exercise adherence in adolescents.

## 2. Materials and Methods

### 2.1. Sample

According to the principle of multistage stratified cluster sampling, the sampling process is divided into several stages. Shanghai has put forward the strategic goal of being a “World-renowned sports metropolis”, which includes playing a representative role in Chinese sports. Using Shanghai as an example, this paper selects four survey areas: Pudong New Area, Putuo District, Minhang District and Yangpu District. Three middle schools were selected in each district and stratified by 7th and 8th grades (first year of junior high school and second year of junior high school). Random cluster sampling was conducted in each grade by class. All students took part voluntarily with written consent from their parents and their own oral consent. After signing the Informed Consent of Investigation, a total of 2200 children and adolescents from 12 schools were selected as investigation objects. A total of 2105 questionnaires were received, and 2087 valid questionnaires were kept after eliminating those with missing items, omissions, or multiple choices. The recovery rate was 96% and the effective recovery rate was 95%.

### 2.2. Measurement

#### 2.2.1. Peer Support

Only four observation variables can meet the latent variables [39], whereas three observation variables can also meet the measurement requirements [7]. The peer support scale of Verloigne et al. [40] was adopted, with 3 items and a 5-level Likert scale. The peer model, peer encouragement and peer accompaniment of the physical activities of children and adolescents were evaluated according to five grades, namely “very consistent (1)~very inconsistent (5)” [8]. The scale question is set as a declarative sentence, using the first-person inquiry method. For example, “My partner often takes part in physical activity”, “My partner often encourages me to take part in physical activity”, “My partner often participates in physical activity with me”. Additionally, reverse scoring is used from 5~1. The weights were decided using the analytic hierarchy process (AHP). They were 41% for peer companionship, 25% for peer encouragement and 34% for peer model. The score of the question item is multiplied by the weight and added to the score for peer support. The higher the score is, the higher the level of peer support will be [9]. The results from the reliability test show that the three questions measured well for internal consistency (Cronbach’s α = 0.89).

#### 2.2.2. Self-Regulation

The self-regulation questionnaire prepared by Stouter et al. [41] was adopted for self-regulation. There are eighteen items in the scale, and the four-point scoring method is adopted, ranging from very inconsistent (1) to very consistent (4). The questionnaire is divided into three dimensions, from the three aspects of selection, optimization, and compensation, to explore the development and problems for all aspects of self-regulation ability. The question is as follows: “When I set a goal, I stick to it”, “I make every effort to achieve a goal”, “For important things, I pay attention to whether I need to devote more time or pay more effort”. The higher the score is, the higher the self-regulating value of subjective cognition will be. The reliability test results show that all the questions have good internal consistency, and the internal consistency reliability coefficients of the three dimensions are 0.75, 0.70 and 0.67, respectively.

#### 2.2.3. Self-Efficacy

Self-efficacy was measured using the general self-efficacy scale (GSES) adapted by Schwarzer [17] et al., based on the maturity scale. Ten items were involved in the scale. The four-point scoring method is used to ask whether subjects would be confident to continue exercising. The questions are as follows: “I can face difficulties calmly because I believe I could handle them”, “Faced with a problem, I can usually come up with several solutions”, and “When in trouble, I can usually figure out ways to cope with it”. At the same time, this scale is a single dimension without a weight problem. The higher the score is, the stronger the general sense of self-efficacy will be. After the localization reliability and validity test, the scale has been widely applied [21]. The results show that the items have good internal consistency (the internal consistency coefficient is 0.87, half reliability is 0.90, retest reliability is 0.83).

#### 2.2.4. Exercise Adherence

Exercise adherence was measured using the exercise adherence scale developed by Wilson and Rodgers [42] et al. The scale consists of 3 items, scored by 7 points, from strongly disagree (1) to strongly agree (7). The given scenarios are as follows: “I will continue to exercise even though I am busy with work”, “I am confident that I will continue to exercise even though I feel tired”, “I will continue to exercise despite the bad weather”. In the meantime, the questionnaire contained a subscale that measured willingness to stick to exercise. Forward scoring is adopted. Similarly, after the weight is determined, the question item score is multiplied by the weight and then added to total score for exercise adherence. The higher the score is, the stronger the exercise adherence will be. The research reliability test results showed that the items had good internal consistency (the total internal consistency coefficient of the scale was 0.94).

### 2.3. Dara Processing

The statistical software SPSS 26.0 was used for data analysis in this study. Firstly, Cronbach’s Alpha was used for reliability verification and confirmatory factor analysis for validity verification. Secondly, after importing the data into SPSS, descriptive statistics were used for demographic analysis. The Pearson correlation coefficient was used to analyze the correlation between peer support, self-efficacy, and self-regulation, and exercise adherence. The process program developed by Hayes was also used for regression analysis and verification of the measured data, as well as the theoretical construction model. Model six was selected and the non-parametric percentile bootstrap method [14] with deviation correction was used to evaluate the significance level of the mediating effect, so as to perform research and analysis.

## 3. Results

### 3.1. Sample Features

A total of 2087 people engaged in this research; males account for 1081 and females 1006, among them, 1068 students were in the 7th grade and 1018 in 8th grade. The specific sample size and demographic analysis are shown in Table 1. In terms of exercise adherence, there is no obvious difference in grade level, but the difference between males and females is significant (*p* = 0.000). Males’ exercise adherence was significantly stronger than females’, indicating that males were more adherent to exercise. For peer support, there was no significant difference in gender, but that of students in the 7th grade was significantly stronger than that of students in the 8th grade (*p* = 0.000), which may be related to lower academic pressure in the 7th grade and more sufficient time and energy to maintain peer relationships. In terms of self-efficacy, the self-efficacy of males is markedly stronger than that of females (*p* = 0.001), but has nothing to do with grade. In terms of self-regulation, there is no notable difference between the genders, but students in 7th grade are obviously stronger than those in the 8th grade (*p* = 0.008). This still can be explained by the fact that students in the 8th grade are under stronger academic pressure than those in 7th grade. Since the 7th grade in middle school comes right after primary school, the curriculum is easier than that of the 8th grade, and the gap in academic grades is smaller, the psychological gap and imbalance are less, and self-regulation is easy to form. Entering the 2nd grade, the increased academic burden easily generates their negative attitude, and lack of confidence and self-regulation.

### 3.2. Descriptive Statistics and Correlation Analysis among Variables

As is shown in the correlation analysis in Table 2, peer support, self-efficacy and exercise (including all dimensions) are positively correlated, while self-regulation is negatively correlated with peer support, self-efficacy and exercise adherence (including all dimensions).

### 3.3. Test of Intermediate Effect

As is shown in Table 3, the SPSS macro program process compiled by Hayes is used to analyze the mediating effect on self-efficacy and self-regulation, regarding the relationship between peer support and exercise adherence under the condition of controlling gender, grade and age. Regression analysis showed that the affected self-efficacy of peer support (β = 0.275, *p* < 0.001) had a direct positive predictive effect on self-regulation (β = −0.357, *p* < 0.001) and a direct negative predictive effect; self-efficacy vs. self-regulation (β = −0.198, *p* < 0.001) has a direct and negative predictive effect; when peer support, self-efficacy and self-regulation simultaneously predicted adolescents’ exercise adherence, peer support and self-efficacy impose a significant positive predictive effect on adolescents’ exercise adherence (β = 0.135, *p* < 0.001; β = 0.493, *p* < 0.001), and self-regulation had a significant negative predictive effect on adolescents’ exercise adherence (β = −0.184, *p* < 0.001).

As shown in Table 4, deviation proofreading non-parametric percentiles were used to further evaluate the mediating effect. The results showed that self-efficacy and self-regulation played a significant mediating effect, whose value were 0.8852. Specifically, the mediating effect was generated through three intermediary chains: firstly, the indirect effect (1) consisted of peer support → self-efficacy → exercise adherence, with an effect size of 0.6321. The confidence interval of the bootstrap at 95% did not contain zero, indicating a notable mediating effect of self-efficacy. Secondly, an indirect effect consisting of peer support → self-regulation → exercise adherence two, with an effect size of 0.169. and the bootstrap at 95% confidence interval did not contain zero, indicating that self-efficacy and self-regulation also played a significant chain-mediating effect between peer support and exercise adherence. Thirdly, the indirect influence had an effect size of 0.084, and the bootstrap at 95% confidence interval did not contain zero, indicating a significant mediating effect of self-regulation. The results showed that the direct effect size of peer support on adolescents’ exercise adherence accounted for 59%; the mediating effect size of self-efficacy between peer support and exercise adherence was 0.632, accounting for 42% of the total. The value of self-regulation between peer support and exercise adherence was 0.169, accounting for 11% of the whole. The mediating effect size of self-efficacy and self-regulation between peer support and exercise adherence was 0.084, accounting for 6% of the total effect. The specific pathways of adolescent peer support on exercise adherence are shown in Figure 2.

## 4. Discussion

The hypothesis has been verified. Correlation and regression analysis showed that the stronger peer support is, the better the adolescents’ physical exercise adherence will be, which is consistent with earlier views. A study based on American adolescents showed that peer support has a greater direct effect than self-efficacy and is the strongest predictor of exercise by adolescents [43]. An important reason for adolescents to participate in and enjoy physical exercise is the sense of belonging, provided by peers and teams [44]. Social integration brings about a sense of belonging to a group with similar interests, concerns and entertainment [45], which parental support cannot provide. Parents can provide an exclusive feeling of attachment between children and parents, but in peer relationships, they can also obtain a sense of emotional intimacy and of security from them [45]. For example, in multi-player sports, such as badminton and table tennis, intimacy leads to confidence of match victory. Some scholars have also mentioned in relevant studies that physical motivation should be considered by teenagers more than that of young and middle-aged people. Adolescents engage in exercise based on interpersonal motivation [28]. Sticking to long-term exercise can effectively improve physical fitness. Adolescents with peer support are willing to do long-term exercise spontaneously after receiving positive feedback, which provides a good insight into improving the physical quality of Chinese adolescents.

Further results show (Figure 2) that peer support influences exercise adherence and it is mediated by three connections.

While peer support affects adolescents’ exercise adherence, self-efficacy imposes a mediating effect. In the relationship between peer support and self-efficacy, peer support can be emotionally supportive to adolescents who participate in physical exercise. Peer relationships have a significantly positive effect on improving self-efficacy. The above can be analyzed by several factors, happiness, attachment and motivation, which can all explain the direct positive effect of peer support on self-efficacy. In physical exercise, encouragement from peers usually comes in the form of praise [46]. In the process of exercise, adolescents enjoy the happiness brought by peer support, and affects their sense of self-efficacy by enhancing subjective happiness [42]. According to Bowlby’s attachment theory, children with a sense of attachment will think that they can depend on others, and this expectation from others provides a basis for the development of self-efficacy [47]. Peer relationships are distinguished by motivation-related factors. Adolescents with strong peer relationships have similar motivations. Meanwhile, by applying cluster analysis, it is found that different combinations of peer relationships (teenagers with strong peer consciousness and those without) will encounter unique sports experiences [48]. In conclusion, peer support contributes to adolescents’ sense of self-efficacy. In the relationship between self-efficacy and exercise adherence, adolescents with higher self-efficacy are more persistent, which is consistent with earlier views. Bandura specifically mentioned self-efficacy in his research [33], as the belief in one’s ability to maintain physical activity in the face of challenges and set-backs, as key to the success of regular exercise. In the social cognitive model of physical activity, self-efficacy (i.e., the belief in one’s ability to lead an active lifestyle) is a major determinant of sustained and health-promoting levels of physical activity [49]. To some extent, self-efficacy can reflect adolescents’ exercise engagement [50] and affect people’s behavior choices and thinking patterns. When peer support affects adolescents’ self-efficacy through the establishment of attachment, happiness and positive motivation, adolescents’ self-efficacy will promote their core beliefs in exercise adherence and life-long exercise [50].

The results showed a notably negative correlation between peer support, self-regulation and exercise adherence. This suggests that peer support has a negative effect on adolescents’ exercise adherence through self-regulation. The reasons for this result are diverse. Self-regulation refers to a physiological process that psychologically enables individuals to direct their own target activities over time and space [32]. However, in some cases, self-regulation may hinder the implementation of certain behaviors [51], which in this paper is shown as hindering the occurrence of exercise adherence. As expressed by Hoyle [52], self-regulation involves efforts or attempts to control or change one’s internal state. Nevertheless, not all self-regulation is explicit or calculated for adolescents [53]. Teenagers are not aware of the positive effects of physical exercise, such as vital capacity, however, once they come across negative emotions caused by physical exercise, negative behaviors would follow. For example, the feeling of peer disparity. Annes [54] also indicated that although physical exercise is the best predictor of weight loss and health maintenance, without the use of proven cognitive behavioral intervention, the adherence to physical exercise is poor. Although effective self-regulation is known as “the cornerstone of healthy mental function”, teenagers are at a special age with sensitive emotions, while affected by the pressure of academic burden and are prone to academic burnout [55] which is manifested as a negative attitude and a lack of confidence. About 30 million Chinese adolescents have mental health problems at different levels [56], which results in psychological stress. Psychological stress [57] is a multi-factor adaptation process that individuals tend to show through their overall psychological and physiological responses when they perceive an imbalance between their needs and their ability to meet them. They choose to escape from the unpleasant internal state, and this power is part of the escape motivation. When adolescents feel the negative emotions caused by exercise, they adopt avoidance goals, that is, continuing to participate in physical exercise on the one hand, and disengagement oriented coping on the other hand [56]. Not all self-regulation, such as stress management, is effective to make people feel better. On the other hand, when peer support influences exercise persistence through self-regulation, and peer support dominates, self-regulation may be weakened. For example, as with conformity psychology, when adolescents meet together, there may be an adolescent who is unwilling to participate in exercise, which leads to his companion giving up exercise. This may also be the cause of this result. However, more studies [57] believe that adopting positive regulation strategies can weaken negative emotions and maintain happiness. In the theory of self-determination, there may be an additional potential need, which is novelty. According to Gonzalez [58], novelty has been shown to be positively correlated with intrinsic motivation in physical education. However, the existing physical education curriculum is monotonous, which leads to adolescents’ lack of interest in participating, and the correct understanding of physical exercise, the willingness to exercise independently and the ability of self-adjustment are not strong.

Adolescent peer support also influences exercise adherence through the chain mediation of self-efficacy and self-regulation. Satisfaction with self-efficacy can explain many of the effects associated with exercise self-regulation [56]. Studies have clarified that emotion regulation self-efficacy is a prerequisite for the use of emotion regulation strategies [59]. Of the social cognitive variables, self-regulation had the largest effect on physical activity. Self-efficacy independent of self-regulation had little effect [49]. The research results indicate that self-efficacy affects exercise adherence by influencing self-regulation, which is consistent with previous studies [60]. Self-regulation is a predictor of changes in self-efficacy [54], but other studies have shown that they hypothesize that social support influences self-regulation of physical activity through self-efficacy, and self-efficacy influences physical activity through self-regulation, but the results did not end up as expected [49]. This may be because social support includes not only peer support, but also teacher support and parental support. At the same time, it is observed that the multi-chain mediating effect is smaller than the independent mediating effect of self-efficacy. Due to the fact that it is part of the same model, self-efficacy has a positive impact on exercise adherence, while self-regulation has a negative impact, and the ambivalence of adolescents is found with high self-esteem and complex inner heart rates. In addition, this study explores the effect of peer support on exercise adherence through a chain mediation of self-efficacy and self-regulation. Although self-efficacy affects self-regulation [61], in the close relationship between self-efficacy and self-regulation, not only does self-efficacy affect self-regulation, but self-regulation also affects self-efficacy [56]. So, peer support is theoretically possible by influencing self-regulation, then on self-efficacy, and finally on exercise persistence.

In summary, the research hypothesis assumed that peer support has an effect on adolescents’ exercise adherence. While peer support can have a positive effect on exercise adherence directly and through the mediation of self-efficacy, the research also found that self-regulation variables can have a negative effect on peer support and exercise adherence, which is slightly different from previous studies. However, it also showed that peer support, self-efficacy, and self-regulation do have an impact on exercise adherence, whether it is positive or negative. Which means that in future education, we should strengthen the guidance on peer relationships and the intervention on their psychological self-efficacy and self-regulation abilities to promote their psychological acceptance of physical activity and enhance the possibility of their conscious participation in exercise, so as to effectively motivate adolescents to engage in physical activity and have better physical health. There are some shortcomings in this study, which need to be perfected in future studies. The target population of this study only considers ordinary teenagers in school, athletes in service and teenagers with physical defects. In addition, peer support can also have an impact on young athletes who have trained professionally for a long time. Studies have found that young athletes who drop out of school have more peer conflicts than those who continued training, and their satisfaction with peer relationship is significantly reduced [62]. People with disabilities also need peer support in sports, but in some cases, they may perceive that they have a physical disability, which can hinder the mental development of the youth with disability. Peer relationships are more complex for them [63]. At the same time, the quality of sports friendships will change in accordance with age, gender, competitiveness, motivation tendency and different stages of friendship [64]. This study only measured adolescents in the 7th grade and the 8th grade, but the effect of peer support will also be affected in line with a change in age, and the effect of peer support on exercise adherence for adolescents of other ages needs further testing.

Finally, the need for peer relationships not only produces peer support, but may also cause other types of peer relationships, like crime among peers [65], and peer victimization [66]. Good peer environments not only have a positive influence, but also have opportunities for negative effects. The study of peer relationships was conducted over 2 years by controlling the amount of time adolescents spent with their friends for 4 weeks. It found that the facts cannot be explained simply by peer support, because the more time adolescents spent with their friends, the more likely they would have conflicts with their friends. As time spent with peers increases, the likelihood of experiencing negative peer interactions also rises [67]. So, it is not enough to just study peer support. It is also necessary to further explore the relationship between other peer relationships and exercise adherence.

This study examines the joint effects of peer support, adolescent self-efficacy and self-regulation, on exercise adherence. Through encouraging different ways for peers to meet to exercise to improve the persistence of independent exercise, starting from the guidance and promotion of adolescent psychology, adolescent mental health problems can be improved on the one hand and adolescent exercise adherence strategy can be improved on the other hand. Indeed, unlike college-level teens, teens’ exercise adherence during middle school is also influenced by the curriculum. Nevertheless, their self-disciplined persistence will have a long-term and profound impact on their future.

## 5. Conclusions

Peer support, which promotes adolescents’ exercise adherence, serves as an external factor, while self-efficacy and self-regulation performs as an internal factor that exerts influence on adolescents’ exercise adherence. At the same time, peer support cannot only impose influence on adolescents in a direct manner, but also affects their self-regulation state through self-efficacy, thus indirectly affecting their exercise adherence. Self-efficacy and self-regulation play a mediating role in the interpersonal effect on exercise adherence.

## Figures and Tables

**Figure 1 children-10-00401-f001:**
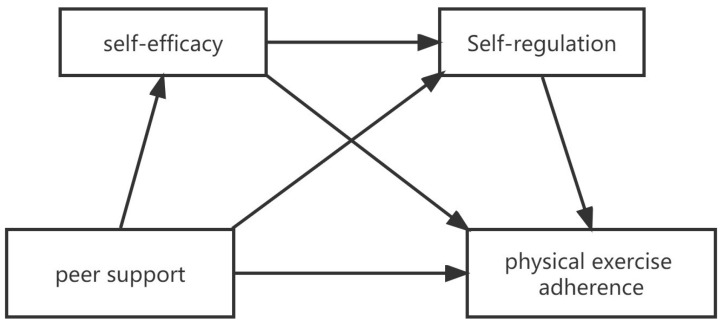
A hypothetical model of peer support affecting exercise adherence.

**Figure 2 children-10-00401-f002:**
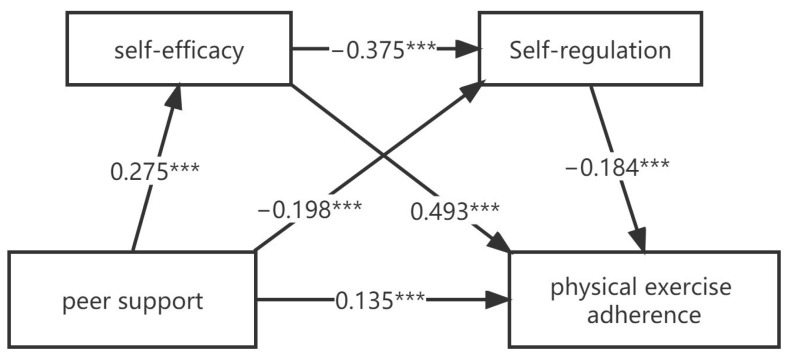
A mediating model of peer support affecting exercise (*** *p* < 0.001).

**Table 1 children-10-00401-t001:** Sample size demographic information and independent sample *t* test.

Dependent Variable	Independent Variable	N = 2087	M ± SD	*t*	*p*
physical exercise adherence	gender	male (1081)	57.47 ± 12.15	3.797	0.000 ***
		female (1006)	55.45 ± 12.24		
	grade	7th grade of junior school (1068)	56.60 ± 12.38	0.377	0.706
		8th grade of junior school (1018)	56.40 ± 12.08		
peer support	gender	male (1081)	12.68 ± 2.78	1.091	0.276
		female (1006)	12.55 ± 2.46		
	grade	7th grade of junior school (1068)	12.83 ± 2.68	3.800	0.000 ***
		8th grade of junior school (1018)	12.39 ± 2.56		
self-efficiency	gender	male (1081)	29.86 ± 6.98	3.325	0.001 **
		female (1006)	28.81 ± 7.40		
	grade	7th grade of junior school (1068)	29.51 ± 7.30	0.999	0.318
		8th grade of junior school (1018)	29.20 ± 7.08		
self-regulation	gender	male (1081)	21.61 ± 9.88	−0.3	0.764
		female (1006)	21.74 ± 9.48		
	grade	7th grade of junior school (1068)	22.22 ± 9.74	2.67	0.008 **
		8th grade of junior school (1018)	21.09 ± 9.59		

** *p* < 0.01, *** *p* < 0.001.

**Table 2 children-10-00401-t002:** Pearson correlation coefficient.

	Peer Support	Self-Regulation	Self-Efficiency	Physical Exercise Adherence
peer support	1			
self-regulation	−0.289 **	1		
self-efficiency	0.276 **	−0.408 **	1	
physical exercise adherence	0.324 **	−0.424 **	0.609 **	1

** *p* < 0.01.

**Table 3 children-10-00401-t003:** Regression analysis between variables.

Equation of Regression		Overall Fit Index		Significance of Regression Coefficient			
Result Variable	Variable of Prediction	*R*	*R2*	*F*	*B*	*t*	*p*
self- efficiency	gender	0.285	0.081	46.120	−0.073	−3.17	0.002 **
	grade				0.042	−3.17	0.327
	age				−0.028	1.453	0.146
	peer support				0.275	13.047	0.000 ***
self- regulation	gender	0.455	0.207	108.680	−0.023	−1.155	0.248
	grade				−0.077	−2.870	0.004 **
	age				−0.007	−0.245	0.807
	peer support				−0.198	−9.698	0.000 ***
	self-efficiency				−0.357	−17.519	0.000 ***
physical exercise adherence	gender	0.652	0.426	256.819	−0.043	−2.565	0.010
	grade				−0.005	−0.230	0.818
	age				0.130	0.578	0.564
	peer support				0.135	7.588	0.000 ***
	self-efficiency				0.493	26.563	0.000 ***
	self-regulation				−0.184	−9.843	0.000 ***

** *p* < 0.01, *** *p* < 0.001.

**Table 4 children-10-00401-t004:** Proportion of the mediating effect.

	Effect Size	BootSE	BootLLCI	BootULCI	Proportion
Total indirect effect	0.885	0.102	0.691	1.088	59%
Ind1: peer support -> self-efficiency -> physical exercise adherence	0.632	0.080	0.486	0.796	42%
Ind2: peer support -> self-regulation -> physical exercise adherence	0.169	0.036	0.105	0.243	11%
Ind3: peer support -> self-efficiency -> self-regulation -> physical exercise adherence	0.084	0.016	0.057	0.120	6%

## Data Availability

The datasets used and/or analyzed during the current study are available from the corresponding author on reasonable request.

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
