# Peer review of "Peer Support and Exercise Adherence in Adolescents: The Chain-Mediated Effects of Self-Efficacy and Self-Regulation"

_children, 2023, doi:10.3390/children10020401_

Round 1

Reviewer 1 Report

Thank you for the opportunity to review this article. The aim of the authors was to investigate the role of peer support on exercise adherence in adolescents. It is understood what the purpose of the study is, however, the authors in several parts of the text miss to highlight what are the results of the research, it would be important to give more emphasis to the content and results of the study, which was conducted on an important topic. The role of exercise should also be emphasized more. There are many typos in the text, the authors were not very consistent in expressing numbers and statistical data.

I have some more comments:

- I suggest that authors add a Keyword related to exercise or physical activity.

- The Introduction is confused. The authors aim to give a description of different dynamics, and how they influence each other. However, I feel that the various paragraphs are poorly connected, and it is difficult to follow the discourse. The choice to separate the various purposes of the study from each other, as well as the hypotheses to be tested, also contributes to the difficulty of understanding what the ultimate purpose of the paper is. In Figure.1, instead of the term "self-efficacy" used in the text, the term "self-efficiency" has been inserted.

- The Measurement paragraph is quite clear

- In Table.1 I encourage the authors to replace the terms "boy" and "girl" with "male" and "female". I also invite them to express the "grade" more concisely, in the current form it is difficult to read.

- I think that Table.2, Table.3 and Table.4 could be eliminated and the content could be expressed discursively. I consider that Figure.2 can also be removed.

- Discussion is rich in information, but at times it is difficult to follow the thread of the discourse. I think the authors should synthesize the text, and better highlight the peculiar aspects of their research, which in this way take a back seat.

Author Response

Dear reviewer:

Thank you very much for your letter dated 02 Jan 2023, and the referees’ reports. Based on your comment and request, we have made extensive modifications to the original manuscript. Here, we attached the revised manuscript in the format of word, for your approval. A document answering every question from the referees was also summarized and enclosed.

Comment 1:it would be important to give more emphasis to the content and results of the study, which was conducted on an important topic. The role of exercise should also be emphasized more.

Response: Thanks for this suggestion. According to your suggestion, I have revised both the introduction and discussion. In the introduction, I added the specific significance of physical exercise affecting adolescents' physical and mental health and emphasized the role of exercise.In the discussion, after analyzing the results, I added a summary paragraph as a way to emphasize the content and results of the study, while re-emphasizing the role of exercise.

Comment 2:There are many typos in the text, the authors were not very consistent in expressing numbers and statistical data.

Response: Thanks for this suggestion. Let me explain the statistical data below. We think the reason why you asked this question is that in the statistics of the number of participants. This study investigated 2200 children and adolescents in 12 schools. Initially, 2,200 questionnaires were distributed, but only 2,105 questionnaires were collected. Then, the questionnaires with missing items, omissions or multiple choices were eliminated, and 2087 valid questionnaires were retained. The questionnaire recovery rate was 96% and the valid recovery rate was 95%, Which may leads to your question, We are not sure whether this explanation makes sense to you, or there are still problems with the data, I would appreciate it if you point it out again. Meanwhile, we also carefully checked and revised typos. We hope to reduce the occurrence of such mistakes. We were sorry for our careless mistakes. Thank you for reminding us.

Comment 3:I suggest that authors add a Keyword related to exercise or physical activity.

Response: I added the keyword: exercise persistence to the article. Based on your suggestion, I added the keyword: exercise persistence to the article.

Comment 4:The Introduction is confused. The authors aim to give a description of different dynamics, and how they influence each other. However, I feel that the various paragraphs are poorly connected, and it is difficult to follow the discourse. The choice to separate the various purposes of the study from each other, as well as the hypotheses to be tested, also contributes to the difficulty of understanding what the ultimate purpose of the paper is.

Response: We thank the reviewer for raising this question. In response to your query, I have made the following changes to the introduction. According to the writing purpose of this paper, the influence of peer support on exercise persistence, the logic of the introduction is reconstructed. The first paragraph describes the current situation of people's exercise today isn't very well. So I wrote the second paragraph to describe the significance of exercise adherence, and I wrote about the research status of persistent exercise in the third paragraph. The fourth paragraph introduces the research based on Basic Psychology Needs Theory. Based on this research, we introduced relatedness, autonomy and competence in Basic Psychology Needs Theory in the following paragraphs. We found that peer support, self-efficacy and self-regulation are closely related to relatedness, autonomy and competence. Therefore, the research hypotheses are introduced and concentrated in the eighth paragraph.

Comment 5:In Figure.1, instead of the term "self-efficacy" used in the text, the term "self-efficiency" has been inserted.

Response: We thank the reviewer for raising this question. I recreated the image in the article and changed the word to "self-efficacy".

Comment 6:In Table.1 I encourage the authors to replace the terms "boy" and "girl" with "male" and "female".

Response: Thank you for reminding me. I replaced "boy" and "girl" with "male" and "female" in the article.

Comment 7: I also invite them to express the "grade" more concisely, in the current form it is difficult to read.

Response: We thank the reviewer for raising this question. Indeed, the word "grade" is ambiguous, and each country has different grade arrangements. So I added brackets after the first occurrence of the word, i.e. in the Sample section, "...by 7th and 8th grades" to explain: (First year of junior high school and Second year of junior high school).

Comment 8: I think that Table.2, Table.3 and Table.4 could be eliminated and the content could be expressed discursively. I consider that Figure.2 can also be removed.

Response: Thanks for raising this question. After taking your advice and reviewing the literature, I think that Tables 2, 3, 4, and Figure 2 cannot be removed. This is because Table 2 is a correlation analysis table, Table 3 is a regression analysis table, and Table 4 is a mediating effect share table.

To test the research hypotheses of this paper, (i.e., whether peer support would have a direct effect on peer support, whether self-efficacy and self-regulation would be mediating effect sizes between peer support and exercise adherence, and the final hypothesis: whether peer relationships, self-efficacy, self-regulation, and exercise adherence constitute a chain mediating effect) this would be achieved through three steps.

The first step is to determine the correlation between peer support, self-efficacy, self-regulation, and exercise adherence; in other words, does exercise adherence have an effect because of the first three variables?

This is what is expressed in Table 2. Once the three variables were determined to be related, data analysis was then conducted to determine whether they had a mediating relationship, which is what is expressed in Table 3. After the first two data analyses, it was concluded that there was a mediating effect between the variables, it was necessary to explore the mediating effect in depth, on the one hand, to test whether the mediating effect was significant, and on the other hand, to test how much of the overall effect was accounted for by each of the mediating effect paths consisting of peer support, self-efficacy, self-regulation, and exercise adherence, i.e., how much of a role these variables had. This is what Table 4 and Figure 2 express. Also Figure 2 visualizes the extent to which each variable influences each other and is a summary of the previous studies and an answer to the research hypothesis. Therefore, I believe that Tables 2, 3, 4 and Figure 2 cannot be removed.

However, this is just our opinion, if we understand the meaning incorrectly or if you think there is a simpler way, please let us know and we will revise it again. Thank you for your suggestions.

Comment 9:Discussion is rich in information, but at times it is difficult to follow the thread of the discourse. I think the authors should synthesize the text, and better highlight the peculiar aspects of their research, which in this way take a back seat.

Response: We thank the reviewer for raising this question. After taking your suggestions, I reorganized the discussion part.

The first to fifth paragraphs discuss the research result, based on the result, we emphasized in sixth paragraph that peer support can directly affect exercise persistence and peer support can indirectly affect exercise persistence through self-efficacy and self-regulation, and there is a chain intermediary effect.

The shortcomings of the study and the way forward are discussed in paragraphs 7 and 8. The last paragraph re-emphasizes the content, results, and significance of the study. At the same time, the specificity of this study, i.e., the reasons for the negative effects of self-regulation, is contended for and explained in detail. These include the sense of falling short brought by peers, academic burnout, psychological stress, misconceptions about physical activity, and the possibility of weakening of self-regulation when peer support dominates the influence, such as herd mentality and novelty.

According to the reviewer's useful suggestions, we improved the manuscript, and marked some changes in red that would not affect the content of the paper. We sincerely thank the reviewers for their enthusiastic work and hope that the revision will be approved. Thank you again for your comments and suggestions.

If you have any questions, please contact us without hesitation.

Best regards!

Reviewer 2 Report

The authors present a strong paper that shows lots of merit. I believe there are a few terminology issues and introduction issues. I commend the authors on this work and I have some suggestions to improve the quality of the paper.

Abstract: In line 16, the authors mention exercise perseverance, which I feel comes out of nowhere. There isn’t mention of this and if they measured this prior. This variable needs to be mentioned prior to giving results on it.

Introduction: Lines 25-30, these first few sentences do not make sense. I cannot understand what the authors are trying to convey to the audience. There is mention of ‘those who already have a program opt out of it in a few weeks…’ what program are they referring to? More clarification is needed

Lines 31-32, why is technology a concern? I would assume it has to do with decrease physical activity, but it is not mentioned here, and it needs to be.

Lines 24-48, I do not believe there is enough clearly presented evidence to justify their purpose of measuring exercise adherence in adolescents. More background information is needed specifically in the physical activity adherence barriers realm in the adolescent population.

Lines 51-52, ‘as they get older…’ as who gets older?

Line 57, the authors mention ‘mid-adolescence’, what age range is this? Definitions of adolescence such as early, mid, late, would be beneficial to understand what age ranges they are referring to.

The authors refer to the basic psychological needs theory (from SDT) as relationships, autonomy, and competence. However, the term ‘relationships’ is incorrect as the correct term should be relatedness. This should be changed throughout the paper. Additionally, the authors refer to self-efficacy, self-regulation, and peer support and use them almost as interchangeable terms for competence, autonomy, and relatedness, which should not be done. They are all different and describe different things. One suggestion would be to explicitly define each term and how they are related and why you chose to use the terms you did. Overall, the introduction needs major revisions as I feel the authors do not fully know how to use all these terms together. This needs to be sorted out in the introduction.

Discussion: the discussion contains topics that are outside the scope of the study such as smoking. This study is focused on physical activity and should remain in this realm.

Overall, there are some issues with terms and variables in the introduction and once those get aligned and teased out, I believe this could be a strong paper. I recommend major revisions to the introduction.

Author Response

Dear reviewer:

Thank you very much for your letter dated 11 Jan 2023, and the referees’ reports. Based on your comment and request, we have made extensive modifications to the original manuscript. Here, we attached the revised manuscript in the format of word, for your approval. A document answering every question from the referees was also summarized and enclosed.

Comment 1:In line 16, the authors mention exercise perseverance, which I feel comes out of nowhere. There isn’t mention of this and if they measured this prior. This variable needs to be mentioned prior to giving results on it.

Response: We thank the reviewer for raising this question. Based on your query, I make the following answer. This article is going to study exercise perseverance, not exercise adherence. The term exercise perseverance was previously used to make the expression more diverse, but misrepresented itself, and has now been removed and replaced with exercise adherence. Thanks again to the reviewers for their careful reading.

Comment 2: Lines 25-30, these first few sentences do not make sense. I cannot understand what the authors are trying to convey to the audience. There is mention of ‘those who already have a program opt out of it in a few weeks…’ what program are they referring to? More clarification is needed

Response: Thank you for your advice. According to your question, I revised this article as follows:

  1. In the first paragraph of the introduction, I changed the order of writing to describe the current situation, that is, people's exercise is not very well nowadays.
  2. I also changed the sentence "have a program opt out of it in a few weeks..." to "many people will give up physical exercise in a short time even if they have an exercise plan", to express that there are many people, even if they have the idea of exercising, will eventually give up physical exercise for various reasons.

Comment 3:Lines 31-32, why is technology a concern? I would assume it has to do with decreased physical activity, but it is not mentioned here, and it needs to be.

Response: We thank the reviewer for raising this question. Therefore, after the sentence "As modern technology continuously develops, this phenomenon is becoming increasingly worrying. So, after the sentence "As modern technology continuously develops, this phenomenon is becoming increasingly worrying:"For example, the emergence of the interesting mobile apps will distract people and weaken their exercise persistence."

Comment 4:Lines 24-48, I do not believe there is enough clearly presented evidence to justify their purpose of measuring exercise adherence in adolescents. More background information is needed specifically in the physical activity adherence barriers realm in the adolescent population.

Response: We thank the reviewer for raising this question. I have described more about the purpose and meaning of exercise adherence. The specific benefits of exercise adherence for adolescents, the current status of the number of adolescents adhering to exercise, and the current status of research on adolescent exercise adherence were added to the introduction, and the role of exercise adherence in awakening people from within, promoting behavior, and maintaining behavior was mentioned in the original article. This paper wants to argue that it is important to study adolescent exercise adherence from these four aspects.

Comment 5:Lines 51-52, ‘as they get older…’ as who gets older?

Response: We thank the reviewer for raising this question. So I changed 'as they get older…', to 'As the Adolescents grow up...'

Comment 6:Line 57, the authors mention ‘mid-adolescence’, what age range is this? Definitions of adolescence, such as early, and mid, would be beneficial to understand what age ranges they are referring to.

Response: Thank you for your suggestions and questions, which made me realize that my expression in this place was indeed unclear. For this reason, I have added parentheses after mid-puberty to specify its age (14-16 years) and added early puberty (11-13 years) because puberty encompasses more than just mid-puberty.

Comment 7:The authors refer to the basic psychological needs theory (from SDT) as relationships, autonomy, and competence. However, the term ‘relationships’ is incorrect, as the correct term should be relatedness.

Response: We thank the reviewer for raising this question. I've changed 'relationships' to 'relatedness' as per the correct term.

Comment 8:Additionally, the authors refer to self-efficacy, self-regulation, and peer support and use them almost as interchangeable terms for competence, autonomy, and relatedness, which should not be done.

Response: We thank the reviewer for raising this question. Seeing your suggestion, I realized that this was indeed a big problem, so I made the following changes to the introductory:

Firstly, I described each term including peer support, self-efficacy, self-regulation, exercise persistence, kinship needs, autonomy needs, and competence needs.

Secondly, I described the correlations between peer relationships and relatedness, self-efficacy and autonomy needs, and self-regulation and competence needs separately: relationship needs to contain many, and peer needs are only one of them, but peer needs including peer support are extremely important for adolescents; self-efficacy is one of the expressions of autonomy needs; and self-regulation is one of the expressions of competence needs.

Finally, it is expressed that even though peer support, self-efficacy, and self-regulation cannot be equated with the three needs, because they are a form of presentation of the three needs, and because motivation occurs when one's needs are met, which in this study means that there will be motivation to exercise, this is how the research hypothesis is formulated.

Comment 8:Discussion: the discussion contains topics that are outside the scope of the study, such as smoking. This study is focused on physical activity and should remain in this realm.

Response: We thank the reviewer for raising this question. I have removed the parts that are beyond the scope of the study, such as smoking.

According to the reviewer's useful suggestions, we improved the manuscript and marked some changes in red that would not affect the content of the paper. We sincerely thank the reviewers for their enthusiastic work and hope that the revision will be approved. Thank you again for your comments and suggestions.

If you have any questions, please contact us without hesitation.

Best regards!

Reviewer 3 Report

Authors presents and interesting and well-designed study focused on correlations and mediating effects of the basic psychological needs on exercise adherence. Results are relevant and interesting so, in this sense, I must recommend the paper for publication.

Nonetheless, I suggest authors to include on discussion section more insights about the complexity of the motivation construct, so that, an alternative explanation for the results on self-regulation in this study could be that when peer support operate as the strongest cause for the exercise adherence (consequence), the feeling of self-control (autonomy) could be minimized and that could create a negative perception on peer supporting contexts. Otherwise, it seems logical that self-efficacy influences physical activity through self-regulation than self-regulation positively affects physical activity through self-efficacy when someone is experiencing stressful feelings or too effort for a particular activity or exercise. Therefore, reality by definition is complex and the correlations explained on the study could partially explain this reality (at most 59% of the explained variance). Then, I will be satisfied if authors consider a similar reflection on their discussion.

Finally, I do encourage authors to include in further studies novelty as a potential basic psychological need that could help for the positive adherence to physical activity and exercise, so there are studies (González-Cutre, D., & Sicilia, Á. (2019), for instance) that point out in this way.

Author Response

Dear reviewer:

Thank you very much for your letter dated 14 Jan 2023, and the referees’ reports. Based on your comment and request, we have made extensive modifications to the original manuscript. Here, we attached the revised manuscript in the formats of word, for your approval. A document answering every question from the referees was also summarized and enclosed.

Comment 1:Nonetheless, I suggest authors to include on discussion section more insights about the complexity of the motivation construct, so that, an alternative explanation for the results on self-regulation in this study could be that when peer support operate as the strongest cause for the exercise adherence (consequence), the feeling of self-control (autonomy) could be minimized and that could create a negative perception on peer supporting contexts.

Response:Thank you for your advice. In this research, the conclusions drawn do differ somewhat from previous studies. Therefore, the interpretation of the results of self-regulation needs to be strengthened. I added a detailed explanation of the reasons that contend for the negative effect on self-regulation in the discussion section. This includes the sense of falling short brought by peers, academic burnout, psychological stress, misconceptions about physical activity, novelty, and the possibility that self-regulation may be weakened when peer support is the dominant influence, such as herd mentality.

Comment 2:it seems logical that self-efficacy influences physical activity through self-regulation than self-regulation positively affects physical activity through self-efficacy when someone is experiencing stressful feelings or too effort for a particular activity or exercise.

Response: We thank the reviewer for raising this question. This is an issue that I had not considered before. After taking your advice and reviewing the literature, I have included a discussion in the second half of the fifth paragraph of the discussion section (introducing the chain mediating role of peer support, self-efficacy, and self-regulation) about how self-regulation may also affect self-efficacy and lead to changes in exercise adherence.

Comment 3:Finally, I do encourage authors to include in further studies novelty as a potential basic psychological need that could help for the positive adherence to physical activity and exercise, so there are studies (González-Cutre, D., & Sicilia, Á. (2019), for instance) that point out in this way.

Response: We thank the reviewer for raising this question. After reading your recommended article, I also found novelty to be an interesting and worthwhile variable to study, but I also found that novelty could also serve as an explanation for the creation of a self-regulatory reverse effect on exercise adherence, so I added that element to my explanation of self-regulation. Thank you again for your suggestion.

According to the reviewer's useful suggestions, we improved the manuscript, and marked some changes in red that would not affect the content of the paper. We sincerely thank the reviewers for their enthusiastic work and hope that the revision will be approved. Thank you again for your comments and suggestions.

If you have any questions, please contact us without hesitate.

Best regards!

Round 2

Reviewer 1 Report

I would like to thank the authors for the effort. I think the changes made to the text have greatly improved its quality. Now you can easily follow the authors' reasoning, and the conclusions are clearly described. I am in favor of publishing the article.

I only have a couple of minor suggestions for authors:

-        At line.58, please replace the word “unique” with “special”

-        Phrase at line 85-87 needs to corrected

-        Please, correct typo at line.424

Reviewer 2 Report

Thank you for addressing the comments.